# Transcriptomic and Metabolic Analyses Reveal the Mechanism of Ethylene Production in Stony Hard Peach Fruit during Cold Storage

**DOI:** 10.3390/ijms222111308

**Published:** 2021-10-20

**Authors:** Yan Wang, Li Deng, Junren Meng, Liang Niu, Lei Pan, Zhenhua Lu, Guochao Cui, Zhiqiang Wang, Wenfang Zeng

**Affiliations:** Zhengzhou Fruit Research Institute, Chinese Academy of Agricultural Sciences, Zhengzhou 450009, China; 15222763167@163.com (Y.W.); 13598072983@163.com (L.D.); 82101186035@caas.cn (J.M.); niucn@126.com (L.N.); Panley@126.com (L.P.); luzhenhua@caas.cn (Z.L.); cuiguochao@caas.cn (G.C.)

**Keywords:** SH peach fruit, cold storage, transcriptome, ethylene, lipid, firmness

## Abstract

Stony hard (SH) peach (*Prunus persica* L. Batsch) fruit does not release ethylene and has very firm and crisp flesh at ripening, both on- and off-tree. Long-term cold storage can induce ethylene production and a serious risk of chilling injury in SH peach fruit; however, the regulatory mechanism underlying ethylene production in stony hard peach is relatively unclear. In this study, we analyzed the phytohormone levels, fruit firmness, transcriptome, and lipidome changes in SH peach ‘Zhongtao 9’ (CP9) during cold storage (4 °C). The expression level of the ethylene biosynthesis gene *PpACS1* and the content of ethylene in SH peach fruit were found to be upregulated during cold storage. A peak in ABA release was observed before the release of ethylene and the genes involved in ABA biosynthesis and degradation, such as zeaxanthin epoxidase (*ZEP*) and 8’-hydroxylase (*CYP707A*) genes, were specifically induced in response to low temperatures. Fruit firmness decreased fairly slowly during the first 20 d of refrigeration, followed by a sharp decline. Furthermore, the expression level of genes encoding cell wall metabolic enzymes, such as polygalacturonase, pectin methylesterase, expansin, galactosidase, and β-galactosidase, were upregulated only upon refrigeration, as correlated with the decrease in fruit firmness. Lipids belonging to 23 sub-classes underwent differential rearrangement during cold storage, especially ceramide (Cer), monoglycosylceramide (CerG1), phosphatidic acid (PA), and diacyglyceride (DG), which may eventually lead to ethylene production. Exogenous PC treatment provoked a higher rate of ethylene production. We suspected that the abnormal metabolism of ABA and cell membrane lipids promotes the production of ethylene under low temperature conditions, causing the fruit to soften. In addition, ERF transcription factors also play an important role in regulating lipid, hormone, and cell wall metabolism during long-term cold storage. Overall, the results of this study give us a deeper understanding of the molecular mechanism of ethylene biosynthesis during the postharvest storage of SH peach fruit under low-temperature conditions.

## 1. Introduction

Cold stress is an important abiotic factor that has serious effects on horticultural plants, decreasing their growth, development, and postharvest maturation. Cold stress can directly affect the fluidity and related enzyme activity of plant cell membranes, resulting in metabolic and material transport disorders, and, ultimately, causing chilling injury (CI) [1]. In addition to changes in membrane lipids, cold stress can also cause abnormal plant cell wall metabolism and changes in hormone levels [2].

In the long-term evolutionary process, plants have developed a series of cold response mechanisms. Transcription factors (TFs), such as abscisic acid (ABA)-responsive element (ABRE)-binding factors (ABFs), and C-repeat binding factors (CBFs), regulate the expression of cold stress-related genes in ABA-dependent and ABA-independent pathways, respectively, to induce the plant cold response. The connections between these two pathways have surfaced in *Arabidopsis thaliana* (L.) Heynh. in recent years. MYB96, a central regulator of ABA-responsive genes, is cold-induced and controls CBF induction for cold adaptation [3]. CBFs can slightly induce the production of endogenous ABA and actively regulate the cold tolerance of plants.

Low temperature storage is widely used to extend the postharvest life of fruit crops, as it reduces the metabolic rate of fruit and inhibits the production of endogenous ethylene [4]; however, long-term refrigeration usually results in a severe decline in fruit quality and results in CI symptoms such as decreased aroma, flavor, and juiciness, increased browning of the peel and pulp, and abnormal postharvest maturation [5].

For example, long-term low temperature conditioning has been shown to decrease the quality of apple (*Malus domestica* Borkh.) fruit by inducing changes in the cellular structure, sugars, lipids, and hormones, as well as altering complex transcriptional regulatory networks involving transcription factors [6]. Disturbance of membrane lipid metabolism under cold stress may lead to browning of the peel and a weakened aroma of refrigerated “Nanguo” pears (*Pyrus communis* L.) [7]. Storing banana (*Musa acuminata* Colla) fruit at temperatures lower than 12 °C usually results in CI, which manifests as peel browning, pitting, and a failure to soften [8].

Peach (*Prunus persica* L. Batsch; genus Prunus, subgenus Amygdalus, family Rosaceae) is an important stony fruit which is popular throughout the world. As peach is a typical climacteric fruit, it is not suitable for storage and transportation, as the ethylene release increases during ripening, resulting in softening [9]. Refrigeration can inhibit the ethylene release rate from peach fruit and delay fruit tissue senescence, thus extending its shelf life, and consequently, market value. However, given its sensitivity to low temperatures, the storage of peach fruit at 0–8 °C for more than 2–3 weeks causes a series of CI symptoms, such as pulp browning, abnormal softening, flocculation, and reduced juice content, which drastically decrease its shelf life and quality [10].

It has been reported that in melting peach fruit, ethylene biosynthesis [2,11,12], lipid metabolism [13], and cell wall metabolism [10] are related to CI. Previous studies have shown that exogenous ethylene affects fruit softening and membrane fluidity by regulating the expression of cell wall- and lipid metabolism-related genes, thus reducing the Cl symptoms in peach fruit [14]; however, the molecular mechanism of ethylene biosynthesis under cold stress and the direct relationship between endogenous ethylene level and fruit softening and lipid metabolism have not yet been explored. Although stony hard (SH) peach fruit usually changes color and contains a high content of soluble solids at ripening, they do not release ethylene, and the mature pulp is very hard and brittle, both on and off the tree [15,16]. In our previous study, peaches during fruit ripening were sampled at four stages—S3, S4 I, S4 II, and S4 III—representing fruits from the end of the second exponential growth phase to the commercial maturity stage. For stony hard peach fruit, S3 represents green fruit background color is green, S4 I represents green/slightly white fruit background color, S4 II represents slightly green/white fruit background color, S4 III represents white fruit background color [17].

Studies have indicated that the low expression level of the flavin monooxygenase gene *PpYUCC11* affects auxin accumulation during SH peach fruit ripening, which inhibits the expression of the ethylene biosynthesis gene ACC synthase *PpACS1* [17]. Additionally, activities of cell wall-modifying enzymes, such as polygalacturonase (PG) and β-galactosidase (β-gal), are low in SH peach fruit [10]. Long-term cold storage is effective in overcoming *PpACS1* inhibition, which induces ethylene production [18], resulting in CI [9,10]; however, the underlying regulatory mechanism remains unclear.

In this study, we aimed to determine the mechanism of ethylene production and CI in SH peach fruit undergoing long-term low temperature storage. We compared changes in hormone levels, fruit firmness, transcriptome, and lipidome in fruit of the SH peach variety ‘CP9’ during cold storage (4 °C). The results of this study provide new ideas for the development of novel fruit preservation technologies and low temperature-resistant varieties.

## 2. Results and Discussion

### 2.1. Changes in Ethylene Release during Storage

SH peach fruit were harvested at the S4 III stage and stored under two different conditions—low temperature (4 °C) and room temperature—for 0 d, 10 d, 20 d, 30 d, and 40 d. Significant differences in ethylene levels were detected between the two storage conditions (Figure 1A): no ethylene was released from SH peach fruit stored at room temperature; however, ethylene release was detected in fruit stored at low temperatures for 20 d, which was accompanied by a decrease in fruit firmness (Figure 1A,B). The amount of ethylene released by fruit stored at 4 °C increased sharply at 40 d (Figure 1A).

### 2.2. Changes in Transcript Abundance during Cold Storage

To understand the molecular basis of CI in SH peach fruit during long-term cold storage, we performed RNA–seq analysis of three biological replicates each of CK0 (stored at room temperature for 0 d), L1 (stored at 4 °C for 20 d), and L2 (stored at 4 °C for 40 d) samples (for a total of 9 RNA-seq libraries). The RNA-seq data generated in this study were deposited in the National Center for Biotechnology Information (NCBI) Sequence Read Archive (SRA) database under the accession number PRJNA699046. After removal of adapters, low quality sequences, and ribosomal RNA reads, approximately 73.38 Gb of cleaned reads were obtained. The RNA-seq reads were mapped to the peach genome, with uniquely mapped reads reaching 92% (Appendix A) and Q30 score > 93% (Appendix A). The RNA-seq data of each sample were highly correlated among the three biological replicates (R^2^ > 0.95; *p* < 0.01) (Appendix A); hence, the accuracy and quality of RNA-seq data were determined to be sufficient for subsequent analyses. Principal component analysis (PCA) showed a clear separation between peach fruit samples exposed to CK and those exposed to low temperatures. PC1 (89%) separated CK and L samples, while PC2 (9.4%) distinguished between L1 and L2 samples (Figure 2A). A total of 2914 DEGs (|Log2FC| ≥ 2; FPKM ≥ 10; *p* < 0.05) were identified among the following three comparison groups: L1 vs CK0, L2 vs CK0, and L2 vs L1. The number of both upregulated and downregulated genes was the highest in the L2 vs. CK0 group (Figure 2B,C), indicating that with the extension of cold storage time, the fruit was affected by more serious transcriptional regulation. A hierarchical cluster analysis (HCA) of the remaining 2914 DEGs was carried out, and the HCA dendrogram revealed 20 subgroups (Appendix A). Venn diagram analysis of the DEGs among the three comparison groups revealed that 70 DEGs were shared between samples stored during long-term cold temperature (Figure 2B). A total of 783 DEGs were significantly differentially expressed in the two pairwise comparisons between cold storage and control at the two time points of L1 and L2 (Figure 2B). KEGG enrichment analysis showed that 2914 DEGs were mainly involved in ‘Biosynthesis of secondary metabolite’, ‘Amino sugar and nucleotide sugar metabolism’, ‘Linoleic acid metabolism’, ‘MAPK signaling pathway—plant’, ‘Galactose metabolism’, ‘Starch and sucrose metabolism’, and ‘Brassinosteroid biosynthesis’ (Figure 2D). DEGs that responded to both short (L1) and long term (L2) cold storage were mainly involved in ‘Biosynthesis of secondary metabolite’, ‘Biotin metabolism’, ‘Anthocyanin biosynthesis’, ‘Flavonoid biosynthesis’, ‘Steroid biosynthesis’, ‘Plant hormone signal transduction’, ‘Starch and sucrose metabolism’, and ‘Fatty acid biosynthesis and metabolism’ (Figure 2E).

### 2.3. Changes in Transcript Abundance during Cold Storage

In addition to participating in the fruit ripening process, plant hormones also mediate stress responses. The expression of genes related to ethylene biosynthesis and signaling was examined, in order to investigate the mechanisms responsible for increased ethylene production in SH peach fruit at 4 °C (Appendix A). Almost all ethylene-related genes were significantly upregulated at 4 °C. Compared with CK0, the transcript levels of ACC oxidase 1 (*ACO1*, *Prupe.3G209900*), ACC oxidase 2 (*ACO2*, *Prupe.4G013800*), ACC synthase 1 (*ACS**1*, *Prupe.2G176900*), and ACC synthase 2 (*ACS2*, *Prupe.5G083500*) were significantly increased in L1, indicating that *ACO1**, ACO2**, ACS**1,* and *ACS2* respond to cold stress initiation in SH peach fruit. A previous study has shown that low expression of the flavin monooxygenase gene (*PpYUCC11*, *Prupe.6G157500*) affects auxin accumulation during SH peach fruit ripening, resulting in the inhibition of the expression of the ethylene biosynthesis gene (*PpACS1**, Prupe.2G176900*) expression [17]. In this study, we found that the increase in *PpACS1* transcript abundance at least partially coincided with ethylene production in SH fruit stored at low temperature; however, no difference was detected in *PpYUCC11* expression during cold temperature storage, suggesting that the biosynthesis of ethylene at low temperatures is independent of the maturation pathway.

At low temperatures, the expression levels of genes involved in ethylene signal transduction (Figure 3) were significantly increased, such as the ethylene response sensor (*ETR2*, *Prupe.1G034300*) and serine/threonine protein kinase (*CTR*, *Prupe.7G117700*). Ethylene-insensitive protein 2 (*EIN2*, *Prupe.6G235600*) expression gradually decreased during refrigeration, which may be caused by the phosphorylation of CTR1. Studies in kiwifruit (*Actinidia spp*.) have shown that ethylene-insensitive protein 3 (EILs) specifically respond to low temperatures and regulated ethylene production by affecting the expression of ethylene biosynthesis genes [19]. In this study, *EIL3* (*Prupe.6G018200*) and EIN3 binding F-box protein (*EBF*, *Prupe.7G244300*) showed a dramatic increase in expression upon refrigeration (Figure 3 and Appendix A). These results suggest that low temperatures promote the transcription of specific genes involved in the ethylene signaling pathway in SH peach fruit, which may affect the biosynthesis of ethylene through a feedback mechanism. However, how low temperatures induce the expression of specific genes in the ethylene signal transduction pathway remains unclear.

ABA participates in plant biotic and abiotic stress pathways. In this study, the ABA content of SH peach fruit peaked at 10 d during cold storage and then decreased (Appendix A). Low temperatures inhibited the release of ABA from SH peach fruit during storage (Appendix A). The expression levels of genes related to ABA biosynthesis and signal transduction were analyzed in the transcriptome data (Appendix A).

ABA accumulation and homeostasis are tightly controlled. Zeaxanthin epoxidase (ZEP) activity can be a limiting factor for ABA biosynthesis in roots [20]. ABA is deactivated by ABA 8’-hydroxylase, which is encoded by *CYP707A*. In this study, we found that the expression of *ZEP* (*Prupe.7G133100*) and abscisic acid 8’-hydroxylase (*CYP707A*, *Prupe.5G013100*) was induced specifically in response to cold stress, and the expression levels of both genes first increased and then decreased during cold storage. Therefore, we hypothesized that these two genes may regulate the ABA content during cold storage. Previous studies have shown that 9-cis-epoxycarotenoid dioxygenase (*NCED*) expression is highly correlated with ABA levels, serving as a key regulatory event in ABA biosynthesis [21]. The expression of NCED3 (*Prupe.**4G**082000*) was induced by low temperatures, which was consistent with the ABA content. Additionally, the expression levels of genes associated with ABA signaling gradually increased during cold storage, including three *PYL* receptor-encoding genes (*Prupe.2G256700*, *Prupe.2G308200*, and *Prupe.5G036800*) and three *SnRK2* genes (*Prupe.5G054500*, *Prupe.7G138200*, and *Prupe.1G573300*) (Figure 3 and Appendix A). In contrast, expression levels of genes encoding negative regulators of ABA such as protein phosphatase 2C (*PP2C*, *Prupe.1G408100* and *PP2C*, *Prupe.3G044900*), gradually decreased during cold storage (Figure 3 and Appendix A). These results indicate that low temperatures affect ABA biosynthesis and signal transduction.

Jasmonic acid (JA) is an oxylipin which, similar to ABA, is also involved in the cold stress response. The transcription factor MYC2—the core regulator of JA signaling—coordinates with ICE1 to participate in the cold resistance pathway [22]. In this study, the expression of *MYC2* (*Prupe.5G035400*) and basic endochitinase B (*ChiB*, *Prupe.8G174900*) genes was upregulated by low temperatures (Figure 3 and Appendix A), indicating that the JA signaling pathway was activated by long-term refrigeration. According to previous studies, MYC2 induced by exogenous ABA treatment participates in ABA-dependent stress response [23]. Exogenous ABA application increases the JA content of plants under salt stress, indicating that JA and ABA coordinate with each other under abiotic stress. In addition to ABA, JA also interacts with ethylene through ERFs during abiotic stress conditions [24]. In the current study, changes in the expression levels of genes related to ethylene, ABA, and JA signaling were observed during cold storage (Figure 3), indicating that the interactions between these three hormones were facilitated during long-term cold storage.

### 2.4. Changes in Transcript Abundance during Cold Storage

Cold stress affects cell wall metabolism, disrupting the natural process of fruit softening and leading to CI symptoms, such as soft pulp and reduced juice content, which damage the texture and affect the shelf life of the fruit. In this study, fruit firmness decreased fairly slowly during the first 20 d of refrigeration, followed by a sharp decline (Figure 2B). A total of 91 DEGs related to cell wall structure and metabolism were identified, indicating that long-term low temperature storage seriously affected the cell wall structure and metabolism. Previous studies have shown that the enzyme activities of PG, expansin (EXP), β-galactosidase, and arabinogalactase are reduced in cold-injured peach fruit [5]. We speculate that the differential expression of genes may ultimately affect the cell wall structure during low temperature storage, resulting in a significant decrease in fruit firmness over time.

To better understand the relationship between the decrease in fruit firmness and ethylene production during refrigeration, we investigated the expression of genes that regulate cell wall structure and metabolism (Figure 4 and Appendix A). Genes, including *PG* (*Prupe.4G116600*), pectin methyltransferase (*PME*, *Prupe.1G006800* and *Prupe.1G529500*), and *EXP* (*Prupe.5G047300* and *Prupe.5g057900*) were induced by low temperatures and were co-expressed with *PpACS1*. Assessment of the expression *GAL* (*Prupe.1G262000*), *β-gal* (*Prupe.6G099700*), and arabinogalactan protein *CAP* (*Prupe.6G101100* and *Prupe.6G195800*) indicated that prolonged exposure to low temperatures induced the transcription of these genes (Figure 4 and Appendix A).

Together, these data suggest that the decrease in fruit firmness after prolonged cold storage is related to the production of ethylene.

### 2.5. Analysis of the Expression of Lipid Metabolic Genes and Changes in Lipid Content

CI is generally caused by damage to the fluidity of plant cell membranes and changes in enzyme activity. Lipids in plant cell membranes play an important role in the maintenance of membrane fluidity and acclimatization to cold stress. During long-term cold storage, the expression of several genes related to fatty acid and lipid metabolism have been shown to vary significantly [25,26,27]. In this study, the genes encoding fatty acid desaturase 2 (*FAD*, *Prupe.6G056100* and *Prupe.7G076500*) and stearoyl-ACP desaturase (*SAD*, *Prupe.8G018700*), as well as genes encoding fatty acid degradation enzymes, such as diacyl-/triacylglycerol lipase (*OBL*, *Prupe.1G434100*), dodecenoyl-CoA isomerase (*DCI*, *Prupe.2G231700*, *Prupe.1G034900* and *Prupe.1G220600*), and 3-ketoacyl-CoA thiolase (*KAT*, *Prupe.1G003300*), were significantly upregulated—whereas those involved in fatty acid biosynthesis were downregulated—during long-term cold storage (Appendix A). Based on the analysis of lipid saturation, it was found that the saturated and monounsaturated lipids increased with time at low temperatures, while the polyunsaturated lipid content increased first and then decreased during refrigeration (Appendix A). Lipid unsaturation affects the fluidity and stability of cell membranes. In this study, long-term cold storage may induce fatty acid oxidation and reduce the content of lipid unsaturated lipids, thereby inducing ethylene production and chilling injury.

Liquid chromatograph mass spectrometry (LC-MS) was used to conduct lipidome analysis, in order to explore the possible relationships between changes in lipid content and ethylene production in peach SH fruit during long-term cold storage. According to the lipidome analysis results, we identified a total of eight groups of lipid molecules belonging to 23 lipid sub-classes and comprising 379 lipid molecules in SH peach fruit during long-term cold storage. Low temperatures induced an overall increase in total lipid content (Figure 5A). The 23 lipid sub-classes included phospholipids (PA, phosphatidic acid; PE, phosphatidylethanols; PS, phosphatidylserine; PC, phosphatidylcholines; PI, phosphatidylinositol; PIP, phosphatidylinositol; PG, phosphatidylglycerol; LPC, lysophosphatidylcholine; LPE, lysophosphatidylethanolamine; and CL, Cardiolipin), sphingolipids (Cer, ceramide; CerG1, monoglycosylceramide; and So, sphingosine), glycerolipids (MGDG, monogalactosyldiacylglycerol; SQDG, sulfoquinovosyldiacyglycerol; MGMG, monogalactosylmonoacylglycerol; and DGDG, digalactosyldiacylglycerol), glyceryl esters (DG, diacyglyceride and TG, triacyglyceride), sterols (AglcSiE, AcylGlcSitosterol ester), coenzyme (Co, coenzyme), and fatty acid fat (WE, wax esters) (Figure 5B). The percentages of sphingolipids (Cer and CerG1), phospholipids (PA, PE, PS, PC, and PG), and glycolipids (DG and TG) were relatively high among all lipids (Figure 5B,C).

In order to maintain normal organelle functions in extreme environments, plants must adjust their cell membrane composition [28]. We observed both the lipid composition and content of SH peach fruit to be altered during cold storage (Appendix A and Figure 5B,C).

During cold storage, lipid content changed significantly, reaching a peak at the L1 stage, while ethylene content reached a peak at the L2 stage. Considering the above, we suspected that prolonged refrigeration may change the lipid composition of the cell membrane, which may eventually lead to ethylene production.

The main lipid component of the plant intima system, sphingolipids, are involved in the regulation of defensin-related programmed cell death (PCD) [29,30]. In this study, the content of Cer increased significantly with the duration of refrigeration time (Figure 5B), suggesting that long-term cold storage of SH peach fruit is potentially accompanied by PCD. Our data also revealed that the content of most saturated and monounsaturated lipids increased, while the number of polyunsaturated bonds significantly decreased during cold storage (Appendix A). These results suggest that the reduction in polyunsaturated sphingolipid content and the increase in saturated and monounsaturated sphingolipid contents in SH peach fruit during cold storage may influence membrane stability and fluidity. Another type of sphingolipid, CerG1, is also an important player in the plant adaptive stress response, where CerG1 species such as CerG1 (d18:2/16:1) and CerG1 (d18:2/16:0) likely play a major role in this process. We observed a decrease in the content of CerG1 decreased during cold storage (Figure 5B), which may have led to a decrease in membrane stability.

Expression levels of genes involved in sphingolipid, phospholipid, and glycerolipid metabolism are summarized in Figure 6 and Appendix A, from which it can be seen that significant changes were observed in the expression of some of these genes. Genes such as sphingolipid ∆8 long-chain base (*LCB*, *Prupe.7G084800* and *Prupe.1G054800*) were induced following exposure to low temperatures (Figure 6), which is consistent with the increase in Cer content. CerG1 is synthesized from Cers by ∆8 sphingolipid long-chain base desaturase (∆*8SLD*) and glucosylceramide synthase (*GCS*), and we found that the expression of ∆*8SLD* (*Prupe.1G411100*) and *GCS* (*Prupe.7G219900*) was induced during cold storage (Figure 6). Studies in Arabidopsis have shown that *∆8SLD* plays an important role in cold resistance [31]. Our data revealed that CerG1 content was affected by the reduction in the expression of ∆*8SLD* in SH peach fruit during cold storage, suggesting that long-term cold storage induces the senescence in SH peach fruit.

The content of phospholipids, including PA, PC, PE, PI, PS, and LPC, decreased in SH peach fruit during long-term cold storage (Figure 5). In addition to de novo biosynthesis from acyl-CoA, different types of PAs are also generated through the phospholipase C/diacylglycerol kinase (PLC/DGK) pathway or phospholipase D (PLD) pathway.

Two genes orthologous to Arabidopsis *PLD1α* (*Prupe.3G084800*) and *PLD ζ* (*Prupe.1G130000* and *Prupe.2G152100*) were upregulated and downregulated, respectively, during cold storage (Figure 6), suggesting that reduction in the PA content of SH peach fruit during long-term cold storage is mainly influenced by the PLD pathway; this is consistent with the results obtained in Arabidopsis [25,32,33]. PA can also be produced from GDPP by lipid phosphate phosphatase (LPP). In this study, the expression of *LPPγ* (*Prupe.6G240200*) was decreased during cold storage (Figure 6), which may at least partly explain the reduction in PA content. Transcript levels of genes related to the PG biosynthesis pathway, such as CDP-diacylglycerol Synthase (*CDS*, *Prupe.4G011700*), PG phosphate synthase (*PGPS*, *Prupe.2G221600*), and PG phosphate phosphatase (*PGPP*, *Prupe.5G063900*), were the highest in L1 samples (Figure 6), and the level of PG increased at low temperatures (Figure 5), indicating that the expression of genes related to PG biosynthesis pathway is sensitive to low temperatures and plays an important role in resistance to long-term cold stress.

It is well-known that the content of glycerides in membranes increases after cold acclimation, which facilitates the formation of lipid bilayers and, thus, reduces the damage to the membrane system. A total of 39 differentially DG species were identified in this study, where more than half of these species were significantly increased in the L1 sample but decreased in the L2 sample (Appendix A). According to the analysis of lipid saturation, polyunsaturation of DG increased significantly in the L1 period, but decreased with the extension of refrigeration time in the L2 period (Appendix A). DG can be produced by the PLC enzyme to hydrolyze PG, PI and PS. It is worth noting that the content of glycerides was positively correlated with *PLC* (*Prupe.2G098700*) mRNA level (Figure 6), suggesting that the transcript level of PLC (Prupe.2G098700) at low temperatures is potentially responsible for the change in DG content. These results suggest that the increase in content and polyunsaturation of DG during the L1 period enhances low temperature resistance, while the destruction of the cell membrane system is potentially responsible for the decrease in DG content and polyunsaturation at the L2 stage.

### 2.6. Key TFs and Their Related Co-Expression Regulatory Network during Long-Term Cold Storage

The ethylene production, fruit softening, and differential lipid rearrangement observed during prolonged cold storage may be caused by the transcriptional regulation of key metabolic enzymes. To reveal the transcriptional regulation of key metabolic enzymes occurring in SH peach fruit during long-term cold storage, we analyzed key TFs and their related co-expression regulatory network. We used the gene expression clustering method to identify genes with similar expression patterns. All of the DEGs were clustered into 20 different profiles (Figure 7A). Genes belonging to profiles 11, 16, and 18 were increased after refrigeration. Analysis of these three profiles using PlantTFDB v5.0 (http://planttfdb.gao-lab.org, 15 August 2021) revealed 203 TFs belonging to 47 different families that showed >4.0-fold difference in mRNA abundance at one or more time points. Of these TF families, the ERF family contained the highest number of DEGs (19), followed by the NAC and C2H2 families, each containing 14 DEGs (Appendix A). If a TF can regulate its gene targets directly, the TF-gene targets probably have a high correlation relationship, and the analysis of TF binding sites (TFBS) between TFs and the promoters of the target genes can reduce the false positive regulatory relationships. In this study, we calculated the Pearson product-moment correlation coefficient (*p* ≥ 0.9) and analyzed the TFBS between different TFs and structural genes using the PlantTFDB v5.0 software (http://planttfdb.gao-lab.org, 15 August 2021) among profiles 11, 16, and 18. According to the above analysis, we constructed a co-expression regulatory network comprising different TFs and structural genes (Figure 7C,D).

A total of 11 TFs belonging to ERF, bZIP, NAC, WRKY, TCP, Trihelix, and C2H2 were positively correlated with ACS1, including four ERFs (Figure 7B). It can be speculated that these transcription factors may be related to ethylene production under cold storage conditions. The ERF transcription factor family had the highest number (19) of differentially expressed genes, and it has been reported to be involved in cold stress responses in Arabidopsis and tomato (*Solanum lycopersicum* L.) [34,35]. With the exception of *Prupe.2G178000*, all other ERFs showed an increase in transcription after refrigeration (Appendix A), suggesting that ERFs play an important role in cold response. Among these ERFs, *Prupe.2G256900* showed the strongest correlation with ethylene, and may regulate the highest number of hormone (ACO, NCED, ZEP, CYP707A, PYL, PP2C, SnRK2, MYC, CHiB), lipid metabolism (PLC, PLD, DGD, ∆8SLD), and cell wall-related genes (PG, PME, Gal, EXP) as well as low temperature-responsive TFs, including NAC, TCP, and WRKY. ERF may alter the hormone levels, firmness, and lipid content in fruit by inducing hormone-, cell wall-, and lipid metabolism-related gene expression under low temperature storage. It can be speculated that the relationships of between ethylene and fruit softening, hormone, and lipid metabolism may be established through the complex regulatory network of *ERF* (*Prupe.2G256900*) TF during long-term cold storage.

CBFs, belonging to the ERFs, are also candidate regulators of SH peach fruit quality during cold storage. The cryogenic response has been shown to be mediated by ICE–CBF-dependent and -independent pathways in plants [36], and increasing evidence has suggested a correlation between these two pathways [37]. The Arabidopsis genome encodes three *CBF/DREB1* genes (*CBF1/DREB1B*, *CBF2/DREB1C*, and *CBF3/DREB1A*), the expression levels of which increase rapidly at approximately 4 °C. According to previous reports, the peach genome contains six *PpCBF* genes. Among these, *PpCBF1* is involved in the cold resistance process of peach fruit [38]. In the current study, the transcript level of *PpCBF1* showed a significant increase in L1 samples, followed by a decline (Figure 7B). In addition, three TFs (LOB, C2H2, and G2-like) and many structural genes related to hormone, lipid, and cell-wall metabolism are positively correlated with *PpCBF1*. *PpCBF1* may bind to the promoters of ABA-related genes, such as *ZEP*, *CYP707A*, and *PYL*, in SH peach fruit, suggesting that *PpCBF1* may participate in the process of cold stress tolerance by inducing ABA biosynthesis and signaling. In Arabidopsis, phospholipids are involved in the CBF regulatory network. Additionally, studies in apple have shown that CBFs bind to the PG gene promoter to induce fruit softening [39,40]. In this study, *PpCBF1* was co-expressed with lipid metabolism-related genes (*TBL*, *Prupe.8G242600* and *PLA*, *Prupe.1G440100*) and a cell wall-related gene (*Gal*). TFBS analysis showed that *PpCBF1* may bind to the C-repeat (CRT)/dehydration-responsive element (DRE) motif in the promoters of these genes, participating in the differential rearrangement of lipids and fruit softening at low temperatures.

The key genes of ethylene biosynthesis (ACS1, Prupe.2G176900) and cold signal regulation (PpCBF1, Prupe.5G090000) were upregulated in SH fruit during refrigeration (Figure 7B), and were found to belong to profile16 (Figure 7A). ACS1 and PpCBF1 did not share the same co-expressed TFs (Figure 7C,D), and ACS1 promoter contains no CBF/DRE motif, indicating that ethylene production during long-term cryogenic storage may be induced mainly through the ICE–CBF independent pathway. According to this network, genes encoding bZIP (Prupe.6G343100), NAC (Prupe.7G001400), WRKY (Prupe.1G393000), and TCP (Prupe.3G252600) TFs bind to the promoter of ACS1, suggesting that ACS1 may be regulated by bZIP, NAC, WRKY, and TCP TFs, all of which are involved in ethylene biosynthesis.

bZIP TFs belong to the AREB/ABF TF family and play an important role in regulating the expression of ABA and stress-related genes. Transgenic Arabidopsis plants overexpressing wheat (*Triticum aestivum* L.) TabZIP60 or TaZIP14-B have exhibited drought, salt, and freezing stress tolerance in an ABA-dependent manner [33]. In this study, the Arabidopsis homolog of bZIP25 (Prupe.6G343100) was identified in refrigerated SH peach fruit. The promoter of ACS1 harbors ABRE. Therefore, we speculate that Prupe.6G343100 regulates ACS1 gene transcription, thus inducing ethylene production.

In addition to bZIP, TFs such as NAC, TCP, and WRKY also participate in abiotic stress response [28,41,42]. NAC, TCP, and WRKY TFs, which regulate ACS1 expression under low temperatures, bind to the promoters of genes related to ABA biosynthesis and signal transduction, lipid metabolism, and cell wall metabolism. Thus, these TFs may regulate the expression of genes related to hormone, cell wall, and lipid metabolism.

### 2.7. PC Promoted Ethylene Production in SH Peach Fruit under Low Temperature

Many studies have many researches have shown that PC is related to the cold resistance in fruit and vegetable of fruit and vegetable [43,44,45]. In this study, the lipid content changed significantly, reaching a peak at the L1 stage, while ethylene content reached a peak at the L2 stage. In order to explore the influence of lipid components on ethylene in SH peach fruit at low temperatures, we soaked CP9 with exogenous PC and stored it at a low temperature. Ethylene production was detectable at 30 d of cold storage, and the ethylene release rate of PC-treated fruits was higher than control when stored at low temperatures for 30 days (Figure 8). PC treatment provoked a higher rate of ethylene production, indicating that lipid composition may affect the ethylene production rate during cold storage. Considering the above, we suspected that prolonged refrigeration may change the lipid composition of the cell membrane, which may eventually lead to ethylene production.

### 2.8. Mechanism and Transcriptional Regulation of of Peach Fruit during Long-Term Cold Storage

The long-term cold storage of fruit (e.g., apples, pears, and bananas) involves a complex multilayered mechanism and transcriptional regulation [6,7]. In this study, the cold storage of SH peach fruit not only stimulated ethylene production and abnormal cell wall metabolism, but also induced changes in ABA content, lipid content, lipid composition, and lipid saturation. During the cold storage of SH peach fruit, ABA content and lipid content increased earlier than ethylene, while firmness began to decrease after ethylene was produced. It has been reported that a burst in ABA concentration preceded that of ethylene production in ‘Plane Sun’ peach fruit, where ABA may trigger ethylene production changes [46]. Ethylene has been shown to be related to lipid and cell wall metabolism under low temperature conditions [2]. PC treatment provoked a higher rate of ethylene production, indicating that the lipid composition may affect the ethylene production rate during cold storage. We speculate that low temperatures may first cause abnormal ABA and lipid metabolism, subsequently causing the production of ethylene which, in turn, promotes a reduction in fruit firmness (Figure 9). It also has been reported that ERF transcription factors may regulate cell wall- and lipid metabolism-related genes in peach fruit under low temperatures. Through transcriptome analysis, we found that ERF may change the hormone levels, firmness, and lipid content in fruit by inducing hormone-, cell wall-, and lipid metabolism-related gene expression under low temperature storage. Our results showed that the long-term cold storage process of peach fruit involves complex transcriptional regulation, including TFs, hormones, cell wall structure, and lipid metabolism pathways (Figure 9).

## 3. Materials and Methods

### 3.1. Plant Material and Treatments

Fruit of the stony hard (SH) peach variety ‘Zhongtao 9’ (CP9) were harvested at the S4 III stage (white fruit background color is white) from the peach breeding nursery of the Zhengzhou Fruit Research Institute, Chinese Academy of Agricultural Sciences (CAAS), China. Pest-free and undamaged peach fruit of uniform size and color were selected and randomly divided into three groups. One group (CK) was stored at room temperature for 30 d (hereafter referred to as CK), while the other group (L) was stored at a low temperature (4 °C) for 40 d. Fruit in both groups were sampled at 0 d, 10 d, 20 d, 30 d, and 40 d after initiating the cold or room temperature storage. At each time point, sample collection was performed in three biological replicates, with each replicate consisting of five individual fruits. After measuring the ethylene release and fruit firmness, the fruit flesh (mesocarp) was cut into small cubes. Then, mesocarp cubes obtained from fruit belonging to the same batch were pooled, frozen immediately in liquid nitrogen, and stored at –80 °C until required for subsequent analyses.

### 3.2. Measurement of Ethylene and ABA Contents and Firmness of Peach Fruit

Ethylene production and flesh firmness were measured as described previously [9]. The ABA content was measured by MetWare (http://www.metware.cn, 1 September 2021) using the AB Sciex QTRAP 6500 LC-MS/MS platform.

### 3.3. Library Construction and Transcriptome Sequencing

A QIAGEN RNeasy Plant Mini Kit (QIAGEN, Dusseldorf, Germany) was used to extract the total RNA from the fleshy peach mesocarp, according to the manufacturer’s instructions. RNA degradation, purity, and integrity were monitored and assessed using electrophoresis in 1% agarose gels, a NanoPhotometer (IMPLEN, Calabasas, CA, USA), and an RNA Nano 6000 Assay Kit with the Bioanalyzer 2100 system (Agilent Technologies, Santa Clara, CA, USA), respectively. First-strand cDNA was synthesized using random hexamer primers and RNase H, and then ligated to adapter sequences with Unique Molecular Identifiers (UMIs). The resulting cDNAs were PCR amplified to obtain the sequencing library. Finally, the library preparations were sequenced on an Illumina platform to generate 150 bp paired-end reads [47].

### 3.4. RNA-Seq Data Analysis

Clean reads were obtained by removing reads containing adapter sequences, poly-Ns, and low-quality reads from the raw data. The Q20 and Q30 scores and GC content of the clean reads were calculated. The UMIs were extracted using the UMI-tools v1.0.0 software [48]. All downstream analyses were based on clean high-quality UMI reads. The peach reference genome and gene model annotation files were downloaded from the Genome Database for Rosaceae (GDR, http://www.rosaceae.org/peach/genome, 10 August 2021). The reference genome index was built using the Hisat2 v2.0 software [48]. Reads were mapped to the reference genome using Hisat2 software, and UMI-tools v1.0.0 [49] was used to deduplicate reads based on the mapping coordinates. HTSeq v0.9.1 was used to count the read numbers mapped to each gene [50]. The FPKM (Fragments Per Kilobase of transcript per Million mapped reads) value of each gene was calculated based on the length of the gene sequence and the number of mapped reads. Genes showing differential expression between two conditions or groups (three biological replicates per condition) were identified using the DESeq R package (1.18.0), based on the following thresholds: |Log2FC| ≥ 2 and *p* < 0.05. The KOBAS software was used to test the statistical enrichment of differentially expressed genes (DEGs) in the Kyoto Encyclopedia of Genes and Genomes (KEGG) database [51].

### 3.5. Lipidome Analysis

Shanghai Applied Protein Technology Co. Ltd. (Shanghai, China) performed the lipidome analysis. Lipid extraction was performed as described previously [41]. The LipidSearch version 4.1 software (Thermo Fisher Scientific Inc., Waltham, MA, USA) was used to identify the raw data for all samples. One-way analysis of variance (ANOVA; *p* < 0.01) was used to analyze changes in lipids between different groups of samples (i.e., stored at 4 °C and at room temperature).

### 3.6. Transcription Factor (TF) Binding Site Prediction and Coexpression Network Construction

Promoter sequences (2000 bp upstream) were retrieved from the peach genome (GDR, http://www.rosaceae.org/peach/genome, 10 August 2021). Then, the PlantTFDB v5.0 software (http://planttfdb.gao-lab.org, 15 August 2021) was used to predict TF binding sites (TFBS) in *A. thaliana* [52]. We constructed coexpression networks between different TFs and structural genes based on the Pearson product–moment correlation coefficient. The correlation coefficient (cutoff of 0.9) was calculated separately, using the expression data under room and cold storage.

### 3.7. Treatment with PC in SH Peach

Approximately 200 ‘CP9’ fruit of uniform size and maturity without wounds or rots were divided into two groups. One group was immersed in 1mM phosphatidylcholines (PC) for 15 min, while the other group was immersed in distilled water for the same time as a control. After air drying, both control and treated groups were stored in baskets at 4 ± 1 °C with 85–90% relative humidity for 30 d. Sampling points were set at 0, 10, 20, and 30 d, and three biological replicates, each consisting of three fruits, were used for ethylene analysis per time point.

## 4. Conclusions

Our results showed that the long-term cold storage process of peach fruit involves complex transcriptional regulation, including TFs, hormones, cell wall structure, and lipid metabolism pathways (Figure 9). During the long-term cold storage of peach fruits, the hormone ABA content was increased, and the lipid composition and content of cell membranes changed. Abnormal metabolism of ABA and cell membrane lipids may promote the production of ethylene under low temperature conditions, causing the fruit to soften. In addition, ERF transcription factors play an important role in regulating lipid, hormone, and cell wall metabolism during long-term cold storage. This study enhances our understanding of the molecular mechanisms underlying ethylene production in SH peach fruit during postharvest storage at low temperatures.

## Figures and Tables

**Figure 1 ijms-22-11308-f001:**
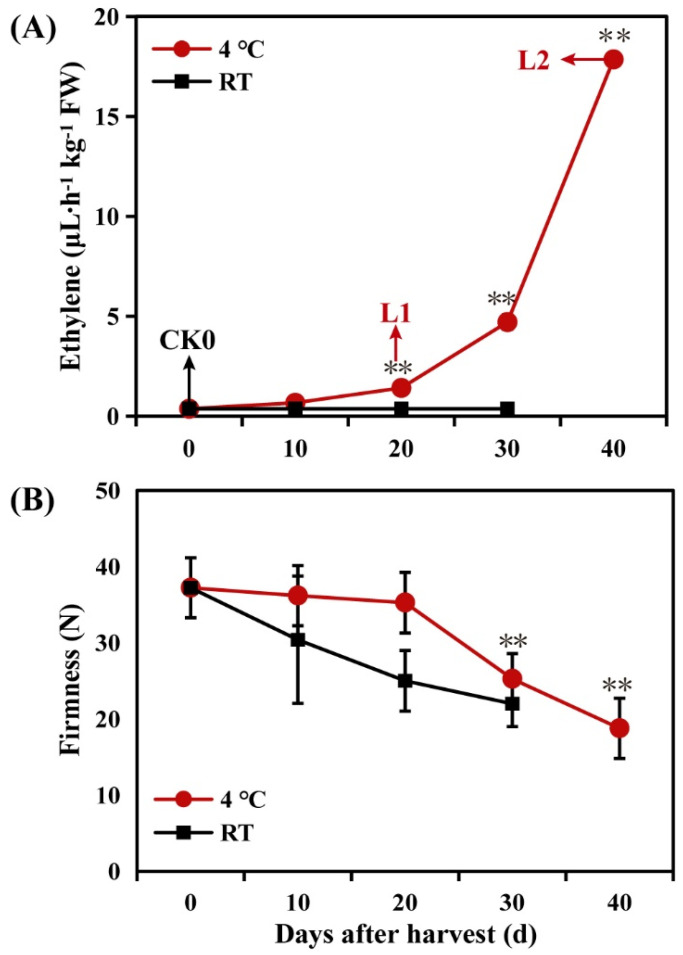
Physiological changes of stony hard (SH) peach fruit stored at room temperature or low temperature: (**A**) Ethylene production rate; and (**B**) Fruit firmness. Arrows indicate the samples used for RNA-seq and lipidome analyses. The asterisks (**) indicate significant differences (*p* < 0.01; test for Student’s *t*-test). The error bars represent the standard error (SE) calculated according to three independent biological replicates. Abbreviations used are as follows: CK0 (stored at room temperature for 0 d), L1 (stored at 4 °C for 20 d), and L2 (stored at 4 °C for 40 d).

**Figure 2 ijms-22-11308-f002:**
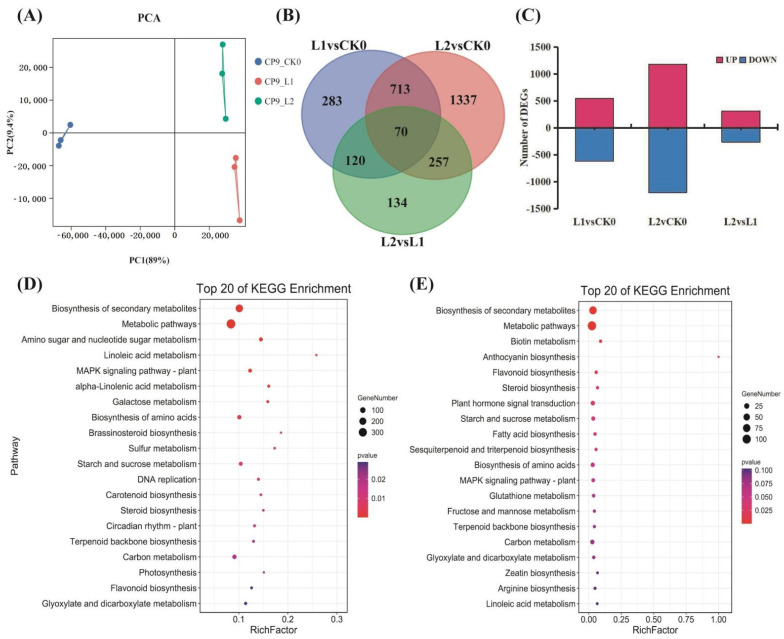
Overview of the RNA–seq data of SH peach fruit stored at room temperature or low temperatures. (**A**) Principal component analysis (PCA) of stony hard (SH) peach fruit during harvest, cold storage, and subsequent ripening. Abbreviations used are as follows: CK0, stored at room temperature for 0 d; L1, stored at 4 °C for 20 d; and L2, stored at 4 °C for 40 d; (**B**) Venn diagrams of DEGs among the different comparison groups (L2 vs CK0, L1 vs CK0, and L2 vs L1); (**C**) Number of DEGs identified by pairwise comparison in the different comparison groups (L2 vs CK0, L1 vs CK0, and L2 vs L1). Up, upregulation; Down, downregulation. Threshold: |Log2FC| ≥ 2, *p* < 0.05; (**D**) KEGG enrichment analysis of 2914 DEGs; (**E**) KEGG enrichment analysis of 783 DEGs.

**Figure 3 ijms-22-11308-f003:**
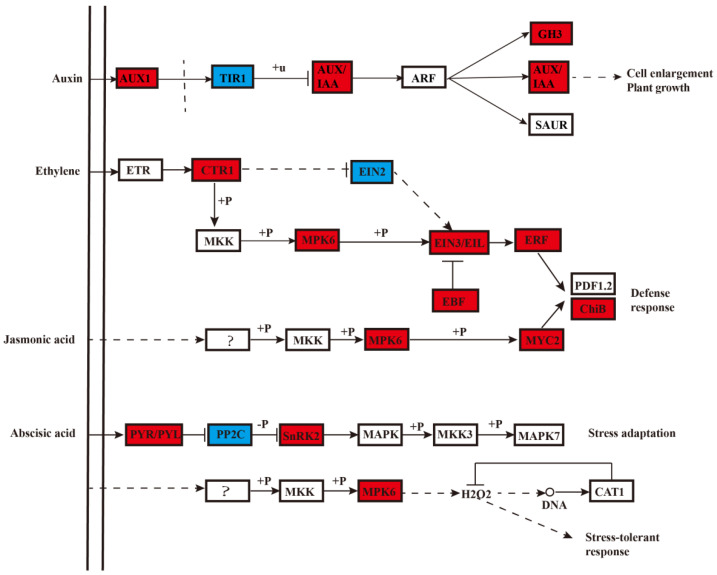
Expression profiles of genes related to hormone signaling. Red boxes indicate genes upregulated after cold storage; blue boxes indicate genes downregulated after cold storage.

**Figure 4 ijms-22-11308-f004:**
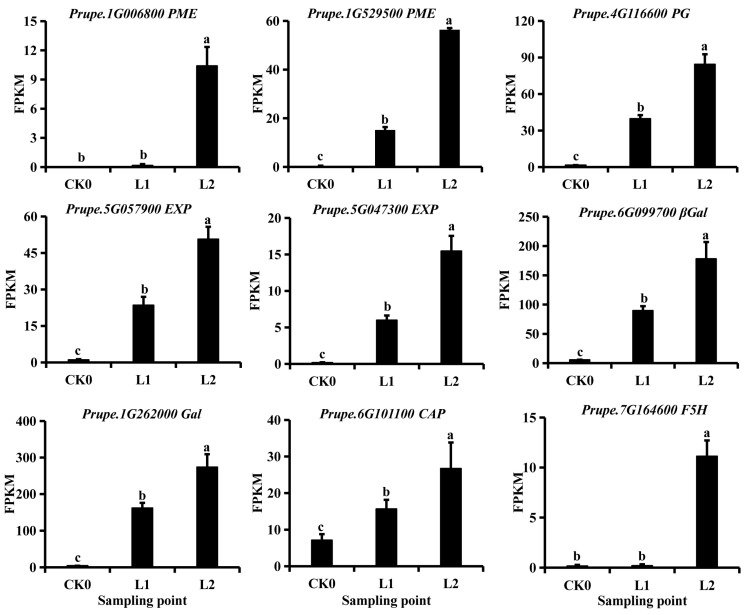
Expression profiles of genes involved in cell wall metabolism. PG, polygalacturonase; PME, pectin methylesterase; Exp, expansin; Gal, galactosidase; -Gal, -galactosidase; F5H, ferulate-5-hydroxylase; CAP, arabinogalactan protein (AGP). The error bars represent the standard error (SE) calculated from three independent biological replicates; in each plot, bars with the lowercase letters (a, b, or c) are significantly different (*p* < 0.05).

**Figure 5 ijms-22-11308-f005:**
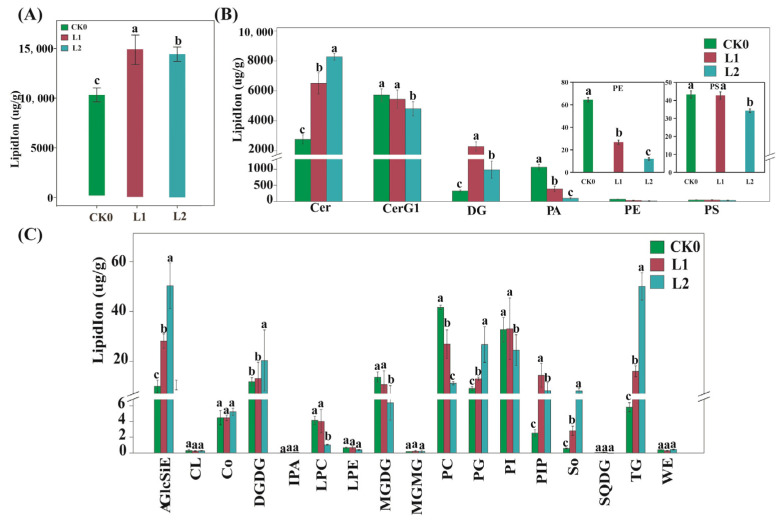
(**A**) Total lipid content of SH peach fruit at the indicated time points during cold storage. (**B**,**C**) Lipid class content of SH peach fruit at the indicated time points during cold storage. LipidIon represents total lipid content. AGlcSiE, AcylGlcSitosterol ester; Cer, ceramide; CerGl, monoglycosylceramide; Co, coenzyme; DG, diacyglyceride; DGDG, digalactosyldiacylglycerol; DGMG, digalactosylmonoacyglycerol; FA, fatty acid; LPC, lysophosphatidylcholine; LPE, lysophosphatidylethanolamine; LPG, lysophosphatidylglycerol; MGDG, monogalactosyldiacylglycerol; MGMG, monogalactosylmonoacylglycerol; PA, phosphatidic acid; PC, phosphatidylcholine; PE, phosphatidylethanolamine; PG, phosphatidylglycerol; PI, phosphatidylinositol; PIP, phosphatidylinositol; PS, phosphatidylserine; So, sphingosine; SQDG, sulfoquinovosyldiacyglycerol; and TG, triacyglyceride. Differences between samples were analyzed by one-way ANOVA, according to Fisher’s least significant difference (LSD) test. The error bars represent the standard error (SE) calculated from three independent biological replicates; in each plot, bars with the lowercase letters (a, b, or c) are significantly different (*p* < 0.05).

**Figure 6 ijms-22-11308-f006:**
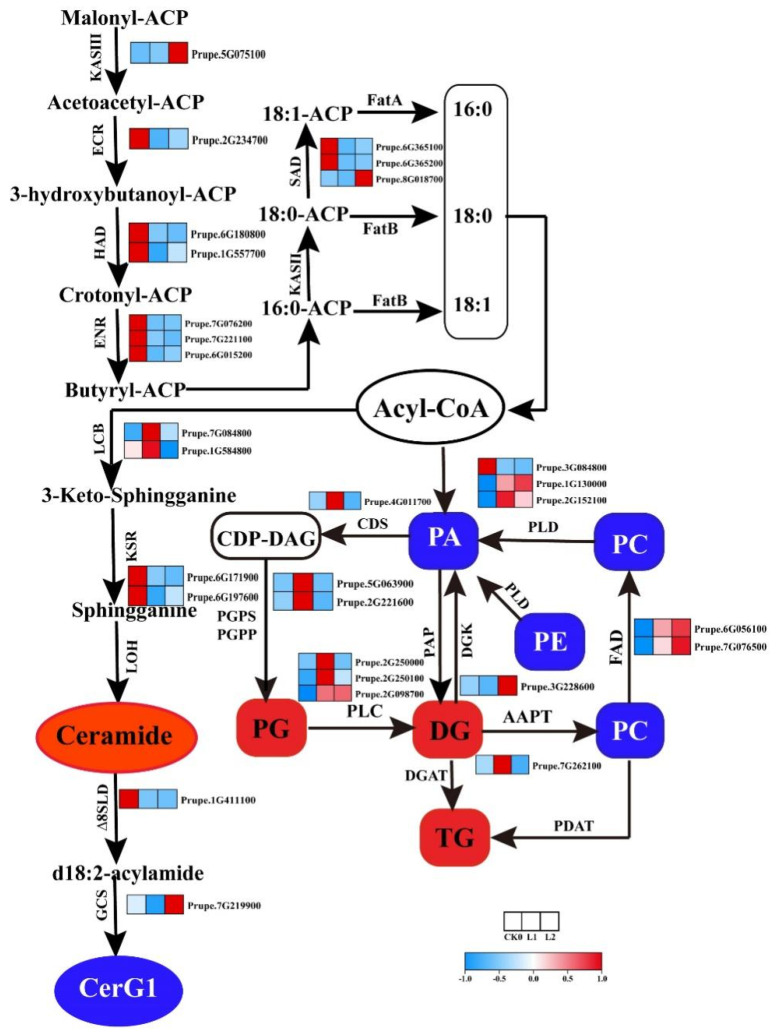
Expression profiles of genes related to lipid metabolism and fatty acid levels in SH peach fruit. Red boxes and ellipse indicate lipid sub-classes showing a significant increase in content after cold storage; blue boxes and ellipse indicate lipid sub-classes showing a significant reduction in content after cold storage. Red and blue boxes in the heatmap indicate upregulation and downregulation of gene expression, respectively.

**Figure 7 ijms-22-11308-f007:**
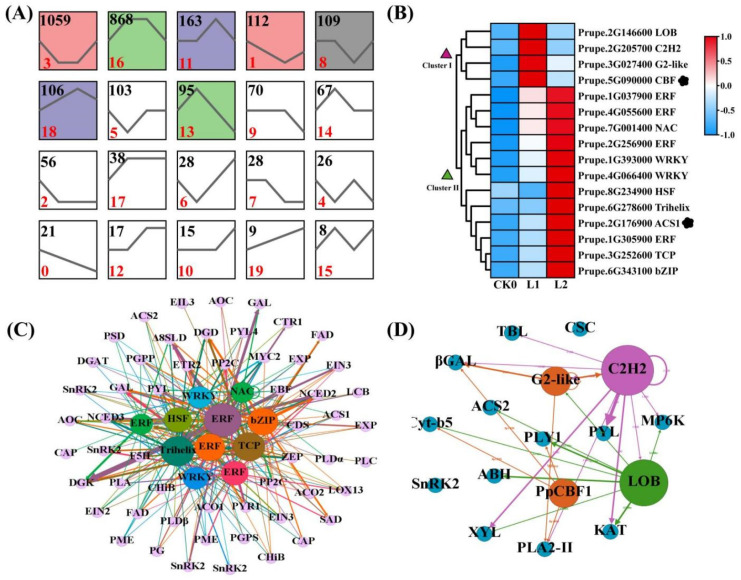
(**A**) Clustering of gene expression data. Expression profiles are based on the cluster of genes differentially expressed at low temperatures. Clusters with colored backgrounds have statistically more assigned DEGs than the number expected. The black line in each box represents the extracted expression pattern for each cluster; (**B**) Heatmap of differentially expressed transcription factor (TF) family members, as well as ACS1 and PpCBF1 genes during SH peach fruit cold storage; (**C**) Co-expression network comprising cluster Ⅱ TFs and genes related to cell wall modification, lipid metabolism, and phytohormone biosynthesis; (**D**) Co-expression network comprising clusterⅠTFs and genes related to cell wall modification, lipid metabolism, and phytohormone biosynthesis.

**Figure 8 ijms-22-11308-f008:**
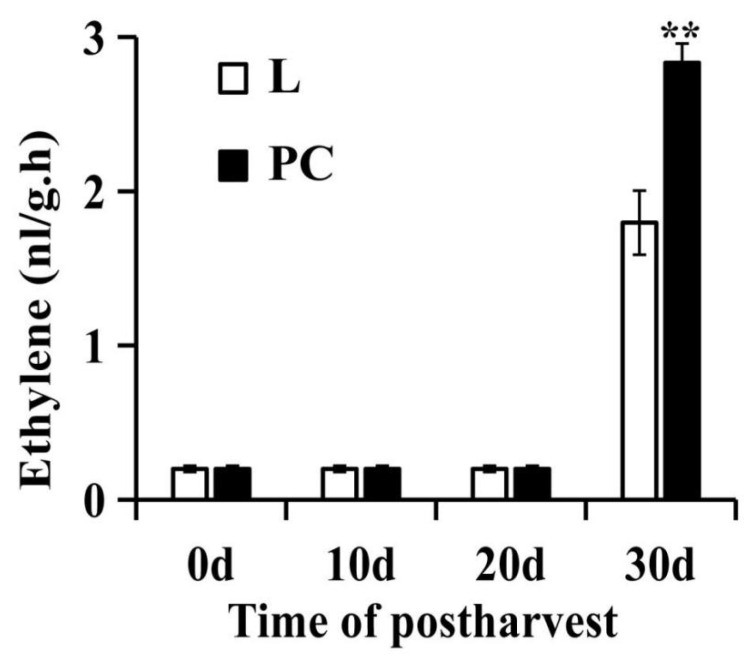
PC promoted ethylene production in SH peach fruit under low temperatures. Error bars represent the standard error (SE) calculated from three independent biological replicates. Asterisks (**) indicate significant differences (*p* < 0.01; Student’s *t*-test).

**Figure 9 ijms-22-11308-f009:**
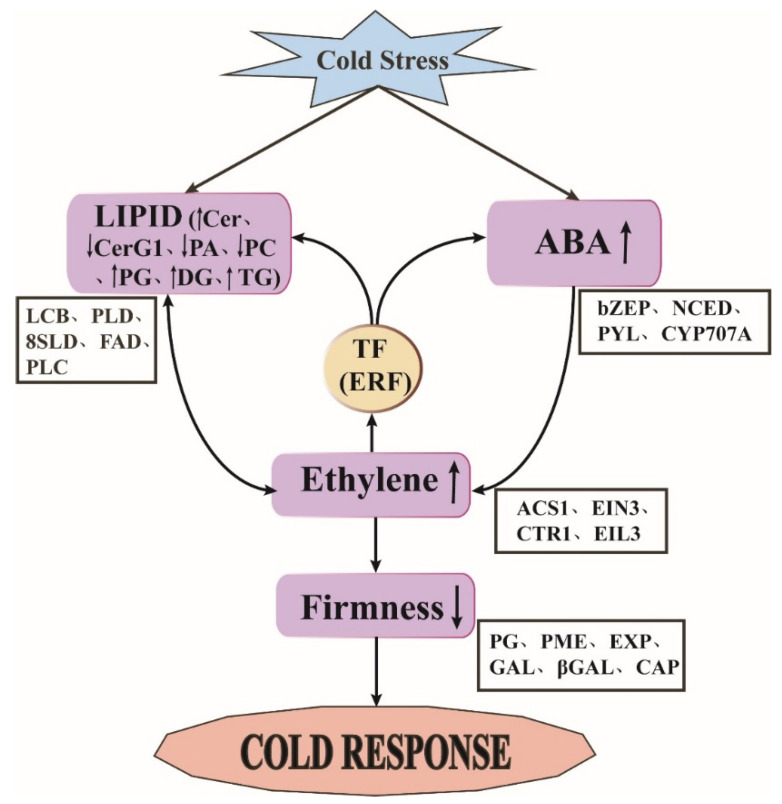
Regulatory model of DEGs involved in pathways related to TFs, phytohormones, cell wall structure, and lipids during cold storage. Arrows indicate changes in ethylene content, lipid content, and fruit firmness.

## Data Availability

The authors declare that the data supporting the study findings can be found in the article and Appendix A or can be obtained by making a request to the corresponding author. The RNA-seq data generated in this study were deposited in the National Center for Biotechnology Information (NCBI) Sequence Read Archive (SRA) database under the accession number PRJNA699046.

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
