# Peer review of "Transcriptomic and Metabolic Analyses Reveal the Mechanism of Ethylene Production in Stony Hard Peach Fruit during Cold Storage"

_ijms, 2021, doi:10.3390/ijms222111308_

Round 1

Reviewer 1 Report

This study shows cold storage might induce the changes of transcriptional factors, hormones, cell structure and lipid metabolism pathways making peach fruits produce ethylene and soften. Lots of factors were investigated. However I think that they should test the SH peach stored at room temperature as a control. Nevertheless we don’t know whether the change of factors tested are dependent on storage time or storage temperature. As shown in Fig.1 the firmness in the peach stored in room temperature seems to start decreasing for 10 days when the wall composition must start to be changed. At that time ethylene was not produced. Thus some changes in the wall at the time may not be related in the ethylene production.

  1. L1 is that stored at low temperature for “30 days”, which is drawn for “20 days” in the whole text.
  2. In the legend of Fig.2: “CK2 stored at room temperature for 10 d” should be omitted.
  3. “2.8” in line 370 should be “2.7”.
  4. About Fig.8: PC is not solved in water, so it must be solved in any organic solvent. In that case the control peach should be rinsed by that organic solvent instead of distilled water.
  5. As shown in Fig.5, PC was decreased according to the storage time. Why the peach was rinsed by exogenous PC?
  6. The white column in Fig.8 may be named “L” instead of “CK”. CK is the group stored in room temperature.

Reviewer 2 Report

The text should be reviewed to correct some missprints (for example lines 398, 431, 437). Figure 9 must be part of the discussion and not the conclusion, so the discussion should be reviewed in order to include it. Include proper separation between figure legends and the text paragraphs.

Round 2

Reviewer 1 Report

I appreciate some your points.

This manuscript is a resubmission of an earlier submission. The following is a list of the peer review reports and author responses from that submission.

Round 1

Reviewer 1 Report

This study shows that the various proteins related to ethylene production are controlled during cold storage in stony hard peach. I think that the transcriptional regulations including TFs, hormones, cell wall structure and lipid pathway were analyzed in details in this study. However I wonder why they looked into CK0, CK2, L1 and L2, which are in various days after harvest. I think that we could compare the effects of cold storage in the same post-harvest days, for example the fruits stored at low temperature for 30days and at room temperature for 30days.

Otherwise we could not know those changes are dependent on the temperature or on the storage time.

Minor points;

  1. Were the data of the fruits in 40 days at the room temperature missing in both (A) and (B)?
  2. What is the conditions of “subsequent ripening” written in the legend of Fig.1?
  3. The axis of ordinate in Fig.1 (B) should be “Firmness”.
  4. The English words in Fig.2 are too small and blurry to read.
  5. The last sentence of the legend in Fi.4 may be “in each plot, bars with the lowercase letter (a, b, or c) are significantly different (p<0.05).”.
  6. Where is the mark “**” in the Fig.5(A)? What were compared in this statistic analysis to? Were the statistic analyses performed also in Fig.5(B)?
  7. The English words in Fig.6 and Fig.7 are too small and blurry to read.
  8. The sentence in L464 should be omitted.

Reviewer 2 Report

The majority of the important details in this manuscript has already been well documented. Low temperature, for example, causes an increase in the expression of ethylene biosynthesis genes or genes involved in ABA biosynthesis.

Although the authors conducted RNA-seq on various storage periods at room temperature vs cold treatment, there was a major flaw in the experiment design.

According to the authors, no ethylene is produced during the entire storage period at room temperature, so the comparison used for the RNA-seq is invalid.

Since peaches are climacteric fruits, the absence of ethylene production at room temperature for the entire 20-day period indicates that the harvest was done too early, rather than at the commercial maturity stage as claimed by the authors.

It is also very questionable how peaches can reach the climacteric stage at 4C!!

Round 2

Reviewer 1 Report

The axis of ordinate in Fig1B has not been changed. I think that authors should draw the reasons why they selected the samples of CK0, CK2, L1 and L2, and why they couldn't test the sample of 40d at room temperature in the text.

Reviewer 2 Report

The authors present RNA-seq data on various storage periods at room temperature vs cold treatment focusing on the regulation mechanism of ethylene biosynthesis in stony hard (SH) peach fruit undergoing long-term cold storage. The work will be of value to those interested in fruit ripening and shelf life. However, at present it is primarily descriptive and does not provide significant insight into the mechanisms by which ethylene influences gene expression nor whether gene expression is directly related to ethylene production under long-term cold storage.

The provided justifications are adding more questions marks than clarifying the raised issues. For instance, the statement “ethylene released by fruit stored at 4 °C increased sharply at 40 d and then decreased due to cell death and other reasons”. How cell death can be triggered by the cease of ethylene production. If the authors refereeing to the mealiness caused by the long-term low temperature, there is no supporting data in this manuscript.  

The experiment design as stands has a major flaw and does not support the authors’ hypothesis.